



# Individual particle compositions and aerosol mixing states at different altitudes over the ocean in East Asia

Kouji Adachi[1*], Atsushi Yoshida[2], Tatsuhiro Mori[3] Nobuhiro Moteki[4], Sho Ohata[5], Kazuyuki Kita[6], Yoshimi Kawai[7], Makoto Koike[8]

[1] Department of Atmosphere, Ocean, and Earth System Modeling Research, Meteorological Research Institute, Tsukuba, Japan
[2] National Institute of Polar Research, Tachikawa, Japan
[3] Department of Applied Chemistry, Faculty of Science and Technology, Keio University, Yokohama, Japan
[4] Department of Chemistry, Graduate School of Science, Tokyo Metropolitan University, Hachioji, Japan
[5] Institute for Space–Earth Environmental Research, Nagoya University, Nagoya, Japan
[6] Department of Earth Science, Graduate School of Science and Engineering, Ibaraki University, Mito, Japan
[7] Research Institute for Global Change, Japan Agency for Marine-Earth Science and Technology, Yokosuka, Japan
[8] Department of Earth and Planetary Science, Graduate School of Science, The University of Tokyo, Tokyo, Japan

*Correspondence to: Kouji Adachi (adachik@mri-jma.go.jp)

**Abstract.** Aerosol composition and mixing state influence its ability to form cloud droplets and ice crystals and to scatter and absorb sunlight, all of which affect its impact on climate. In this study, aerosol samples were collected over the ocean in the western North Pacific at different altitudes from the sea surface to ~8000 m using an aircraft and a research vessel in the summer of 2022. During the campaign, we had samples originating over the ocean, desert, Siberian Forest biomass burning event, and other sources in East Asia. These samples were classified into three periods based on the sampled air parcel sources and particle compositions measured using transmission electron microscopy. Samples from period 1 had high sea salt and mineral dust fractions, while in period 2, the samples had high fractions of potassium-bearing particles with organics and black carbon concentrations, indicating that they originated from a Siberian Forest biomass burning event. The samples from period 3 showed influences of both sea spray and biomass burning. The number fractions of aerosol types also varied depending on particle size and sampling altitude. Compositions of biomass burning and sea spray were mixed at individual particles and the extent of their mixtures depended on the sampling periods and altitudes. Our results showed a wide range of particle compositions and mixing states, which vary with aerosol source, size, and altitude. These factors need to be considered when evaluating aerosol composition and mixing state, both of which affect aerosol climate effects.





## 1. Introduction

Atmospheric aerosols have a major impact on climate by scattering and absorbing sunlight and becoming condensation cloud nuclei (CCN) and ice nucleating particles (INPs) (e.g., Pöschl, 2005; Pósfai and Buseck, 2010). They also affect human health and visibility. These impacts depend on their composition, shape, and physical and chemical configurations at individual scale (mixing states) in addition to their size distribution and concentration in the atmosphere (Li et al., 2024; Reid et al., 2018). These aerosol properties depend on their emission sources, such as anthropogenic, marine, desert, forest, and biomass burning (e.g., Carslaw et al., 2010; Satheesh and Moorthy, 2005), as well as atmospheric processes.

Aerosol particles can transport for days or weeks in the atmosphere from their emission sources and are eventually removed from the atmosphere by precipitation and dry deposition (e.g., Gao et al., 2022; Oshima et al., 2012). As they travel through the atmosphere, they change their original composition, size, and shape by interacting with gases and other aerosol constituents (e.g., Moteki et al., 2007; Riemer et al., 2019). Eventually, they form mixed particles that contain several or more components originating from different sources within individual particles (Adachi and Buseck, 2008; Li et al., 2016; Riemer et al., 2019). Such particle mixing states affect the radiative forcing of aerosols, thereby altering their response to climate (Adachi et al., 2010; Cappa et al., 2012; Chung and Seinfeld, 2002; Jacobson, 2001). Therefore, the physical and chemical properties of aerosol particles from different sources and transport histories need to be clarified to accurately evaluate their climate impact (Pöschl, 2005).

Asia is a densely populated region and its anthropogenic aerosol emissions have a significant impact on the global scale (e.g., Hoesly et al., 2018). In addition, Asian dust events from, e.g., the Gobi and Taklamakan deserts, cause regional pollution events over East Asia (Uno et al., 2009; Wang et al., 2008). Biomass burning events, including the Siberian Forest fires, can also become significant aerosol sources and affect regional air quality (Agarwal et al., 2010; Johnson et al., 2021; Matsui et al., 2013; Warneke et al., 2010). As a result, East Asia has a complex mixture of aerosols from various sources (Zhou et al., 2018), and these aerosols have a large impact on global climate (e.g., Hoesly et al., 2018; Yang et al., 2024).

In this study, we conducted simultaneous airborne and shipboard observations (Koike et al., 2025). Airborne observations using an aircraft can measure atmospheric gases and aerosols at different altitudes and across a large area in a single day. Shipboard measurements can continuously observe the air for weeks and have a large payload. The airborne observations conduct measurements in both the boundary layer and the free troposphere, whereas the shipboard observations measure the boundary layer, which is strongly influenced by the ocean surface. The simultaneous airborne and shipboard observation has the advantage of collecting samples from the sea surface to high altitudes and can cover a wide range of area and time scale over the ocean, which is one of the largest natural global aerosol sources, emitting aerosols through sea spray and biological activity (Lewis and Schwartz 2004; Vignati et al., 2010).

We collected aerosol samples in the western North Pacific over the ocean near Japan using an aircraft and a research vessel. The sampling allows us to evaluate the mixing of sea spray and long-range transported aerosol particles over the ocean. We used transmission electron microscopy (TEM), which is a powerful tool to measure the compositions of small particles



and their mixing states at an individual particle scale. This study aims to characterize individual aerosol particles from different altitudes in the East Asia and evaluate the implications for their climate impacts.

## 2.    Materials and methods

### 2.1 Sampling campaign using an aircraft and a research vessel

The airborne observation was conducted as part of the Aerosol Radiative Forcing in East Asia (A-Force 2022)
campaign using a Beechcraft King Air, twin-turboprop aircraft (Koike et al., 2025). This aircraft was based at Memanbetsu Airport in Hokkaido between July 19 and August 2, 2022, and conducted eight research flights and two transit flights. This study focused on five research flights on July 22, 27, 29, and 30 and August 1, 2022. Based on Table 3 in Koike et al. (2025), these flights are referred to as flight 4 (abbreviated as F4), F6, F8, F9, and F10, respectively. These selected flights included simultaneous aerosol sampling from both the aircraft and the research vessel during the periods of different atmospheric events
(Fig. 1). Each flight lasted about 4 to 5 hours, starting at about 10 am local time (Japanese standard time; or 1 am UTC). Filter-based aerosol composition measurements using a complex amplitude sensor (CAS) and a scanning electron microscope during the same flights were presented by Ohata et al. (2025), who showed the number concentrations and size distributions of mineral dust particles.

The research vessel (*R/V Shinsei-maru*) departed from Yokosuka on 15 July and had continuous observations from
18 July to 1 August at the western North Pacific collaborating with the aircraft (JURCAOS and JAMSTEC, 2022; Koike et al., 2025) (Fig. 1). Both aircraft and research vessel had synchronized observations for ~20 min to 1 hour for every research flight day. Details of the airborne and shipboard measurements were also described in a review by Koike et al. (2025). Other results obtained during this campaign have been reported elsewhere (Moteki et al., 2023; Ohata et al., 2025; Yoshida et al., 2024).

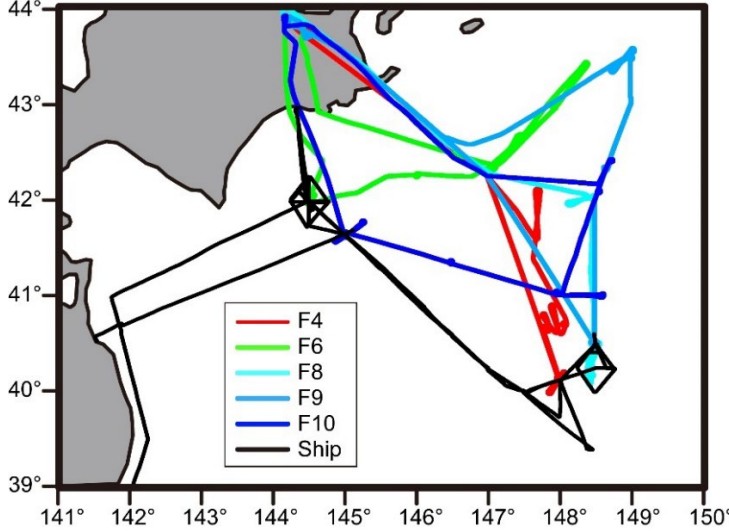

Figure 1. Trajectories of airborne and shipboard measurements used in this study.



## 2.2 Aerosol samplings during the airborne and shipboard measurements

Aerosol samples were collected on 200 mesh TEM grids with Formvar substrate using impactor samplers (AS-24W, Arios, Japan) during both airborne and shipboard sampling. The samplers collected fine-mode aerosol samples with 50% lower/upper cutoff aerodynamic diameters of 300 and 700 nm, respectively, using an airflow of 1 L min$^{-1}$. The samplers used

disks equipped with 24 TEM grids. They collected samples by automatically changing TEM grids with a timer setting of sample collection time. Details of the TEM sampler were described elsewhere (Adachi et al., 2021, 2022).

The airborne sampling used a forward-facing isokinetic inlet (Droplet Measurement Technologies, USA) (McNaughton et al., 2007). During each flight, 24 TEM samples with 11 min sampling time were collected at 1 min intervals (e.g., 00:00-00:11; 00:12-00:23; 00:24-00:35...). The sampling time is the same as the previous airborne measurements in this

area (Adachi et al., 2022) and was sufficient to cover the entire flight time (~4-5 hour) with adequate particle loading.

For the shipboard sampling, samples were continuously collected from an inlet at 14 m above sea level using 30 min sampling times with 90 min intervals (e.g., 01:00-01:30; 03:00-03:30; 05:00-05:30...) (Koike et al., 2025). The sampling time was chosen to collect enough particles under clean conditions (e.g., Adachi et al., 2023). The disks containing 24 TEM grids for the shipboard sampling were replaced every 48 hours.

## 2.3 Transmission electron microscopy measurements

Aerosol particles were measured using a transmission electron microscope (JEM-1400, JEOL, Japan) equipped with an energy dispersive X-ray spectrometer (EDS; X-Max80, Oxford Instruments, Japan) for both TEM and scanning transmission electron microscopy (STEM) modes. The TEM and STEM modes were used for taking TEM imaging and for compositional measurements, respectively. First, we took ~30 TEM images from all samples (198 airborne samples and 153

shipboard samples). Second, we selected one representative area per TEM sample based on the TEM image analyses for the five flights (109 TEM samples) and 13 shipboard TEM samples that covered the synchronized measurements with the aircraft (two or three shipboard samples during each flight) (Table 1). Third, we measured all particles (89 particles on average) within the selected area of 6,000 × (~220 μm$^2$) using STEM with EDS measurements (STEM-EDS). For the airborne and shipboard samples, we measured 9,058 and 1,650 particles from 109 and 13 TEM grids, respectively (Table 1).

Using STEM-EDS measurements, we analyzed particle compositions, sizes, and shapes. Within each area, we selected a threshold to discriminate between particles and substrate in the dark-field STEM image (Adachi et al., 2019). Here, we measured all detected particles with an area-equivalent diameter greater than 0.19 μm, which is equivalent to 100 pixels in the STEM images. An area-equivalent diameter is the diameter of a sphere with the same area as the projected area of the selected particle and typically indicates larger values than that of the aerodynamic diameter of the same particle when a particle

spreads over the substrate. The STEM image, area equivalent diameter, and EDS spectrum were obtained for each individual particle. Elemental compositions (weight percent) within selected elements (C, N, O, Na, Mg, Al, Si, P, S, Cl, K, Ca, Ti, V, Cr, Mn, Fe, Zn, and Pb) were obtained from EDS spectra using 120 kV acceleration voltage and 20 s aquation times. Detection



limits of selected elements were typically 0.02 wt% based on one sigma of the measured peak intensity. TEM images were taken before and after the STEM-EDS measurements to measure the mixing states and detect inclusions within the particles (Figs. S1 and S2). The conditions of the STEM-EDS measurement are the same as those used in our previous studies (e.g., Adachi et al., 2023, 2025).

Table 1. Summary of samples used in this study.

|  | Period | Flight # | Sampling date | TEM sample # | Particle # | Possible sources detected in this study |
|---|---|---|---|---|---|---|
| Aircraft | 1 | Flight 4 (F4) | 22 July, 2022 | 22 | 1050 | Sea spray, Desert, Biomass burning |
|  |  | Flight 6 (F6) | 27 July, 2022 | 22 | 2032 |  |
|  | 2 | Flight 8 (F8) | 29 July, 2022 | 20 | 1975 | Biomass burning |
|  |  | Flight 9 (F9) | 30 July, 2022 | 23 | 2121 |  |
|  | 3 | Flight 10 (F10) | 1 August, 2022 | 22 | 1880 | Sea spray, Biomass burning |
| Ship | 1, 2, 3 |  | 22 July to 1 August | 13 | 1650 | Sea spray, Biomass burning |
| Total |  |  |  | 122 | 10708 |  |

## 2.4 Particle classifications

Based on the STEM-EDS measurements and threshold weight percent values of tracer elements, we classified the measured particles into seven categories (Fig. S3). The particle types and tracer elements are as follows: (1) Mineral dust particles (Al and Si), (2) sea salt particles (Na and Mg), (3) K-bearing particles (K), (4) Ca-Mg-bearing particles (Ca and Mg), (5) sulfate (S), (6) carbonaceous particles (C and O), and (7) others. The particle images and compositions were checked one by one to verify that the categorized particles reasonably represented the particles in the samples.

Although individual particles are mixtures of several components (e.g., sea salt + sulfate particles), we have simply classified them into these single categories, focusing mainly on primary aerosol particles. Such aerosol species include mineral dust and sea salt particles, which are important INP and CCN contributors to the activation of ice crystals and water droplets, respectively. Meanwhile, it should be noted that the number classification differs from that based on aerosol mass concentrations. That is, secondary aerosol particles (e.g., sulfate and organic aerosol particles) that are condensed onto pre-existing particles and are detected from nearly all particles, will be underestimated in the number fractions when compared to their mass fraction values.

## 2.5 Black carbon measurements

A Single Particle Soot Photometer (SP2, Droplet Measurement Technologies, USA) was used to measure the mass concentrations of black carbon (BC) particles from the airborne measurements. The BC mass concentration in this study was based on those with a mass equivalent diameter of 0.071 - 1.0 μm by assuming a void-free density of 1.8 g cm$^{-3}$ (Ohata et al., 2025). The conditions of the SP2 measurements were described in the previous reports (Koike et al., 2025; Ohata et al., 2025). Details of this instrument are also described in Moteki and Kondo (2010).



**2.6 Back trajectory measurements**

We used a Meteorological Data Explorer (METEX) back-trajectory model provided by the National Institute of Environmental Studies (NIES) (Zeng et al., 2003) to evaluate the possible source of an air parcel sampled by the aircraft (Figs. 2, S4, and S5). We also plotted fire spots that occurred during the campaign using NASA's Fire Information for Resource Management System (NASA FIRMS, 2025).

**3.    Results**

**3.1 Possible sources of the sampled air parcel**

     During the campaign, the sampled aerosol particles originated from or were transported over the ocean, desert, Siberian Forest biomass burning, and other sources (e.g., anthropogenic and small biomass burning sources) (Fig. 2). We characterized each flight based on the aerosol sources using a five-day back trajectory model starting at hourly aircraft sampling positions from 2:00 to 6:00 UTC (Figs. 2, S4, and S5). The research flights were grouped into three periods based on their 150 trajectories. On July 22 and 27 (F4 and F6, respectively), the air parcel came from the ocean around Japan and the land over northern East Asia (period 1). On July 29 and 30 (F8 and F9, respectively), the air parcel came mainly from the northwest over large biomass burning events in the eastern Siberian Forest and around northern East Asia (period 2). On August 1 (F10), the air parcels originated from the ocean northeast of Japan and the Siberian Forest biomass burning area (period 3). The back trajectories of all flights were mostly transported from below ~3000 m, suggesting that they could be influenced by ground-155 level emissions such as biomass burning and sea spray (Fig. S4).

     BC is a tracer of combustion, including biomass burning and anthropogenic sources (Bond et al., 2013). BC mass concentrations were measured at different altitudes for each flight. BC concentrations showed an increase at altitudes between 1000 and 3000 m, peaking at ~2000 m for all flights (Fig. S6). The flights during period 2 (F8 and F9) had the highest BC concentrations, peaking at ~400 ng m$^{-3}$ and ~1000 ng m$^{-3}$, respectively. The trajectories of both flights originated from Siberian 160 Forest biomass burning. The BC concentration on period 3 showed some increase (~100 ng m$^{-3}$) when the air parcel originated from the biomass burning area. In contrast, the BC concentrations during period 1 were relatively low (mainly < 100 ng m$^{-3}$). The measurements of BC concentrations together with the trajectory analysis suggested a strong influence of Siberian Forest biomass burning during the campaign, especially for period 2. This interpretation is supported by the studies of Koike et al. (2025) and Ohata et al. (2025), who showed the influences of biomass burning based on their observations and modelling. The 165 altitude at which the BC concentration peaked (~2000 m) was just above a cloud layer (1700 -1900 m), suggesting that BC below the cloud layer may be removed by the cloud and precipitation (Koike et al., 2025). Although anthropogenic sources and small-scale biomass burning in northern East Asia (< ~50° N) emit BC as seen in the measurements during period 1 (Fig. 2), their contributions during our measurements could be smaller than that from Siberian Forest biomass burning as samples from period 2 had much higher BC concentrations than other periods.



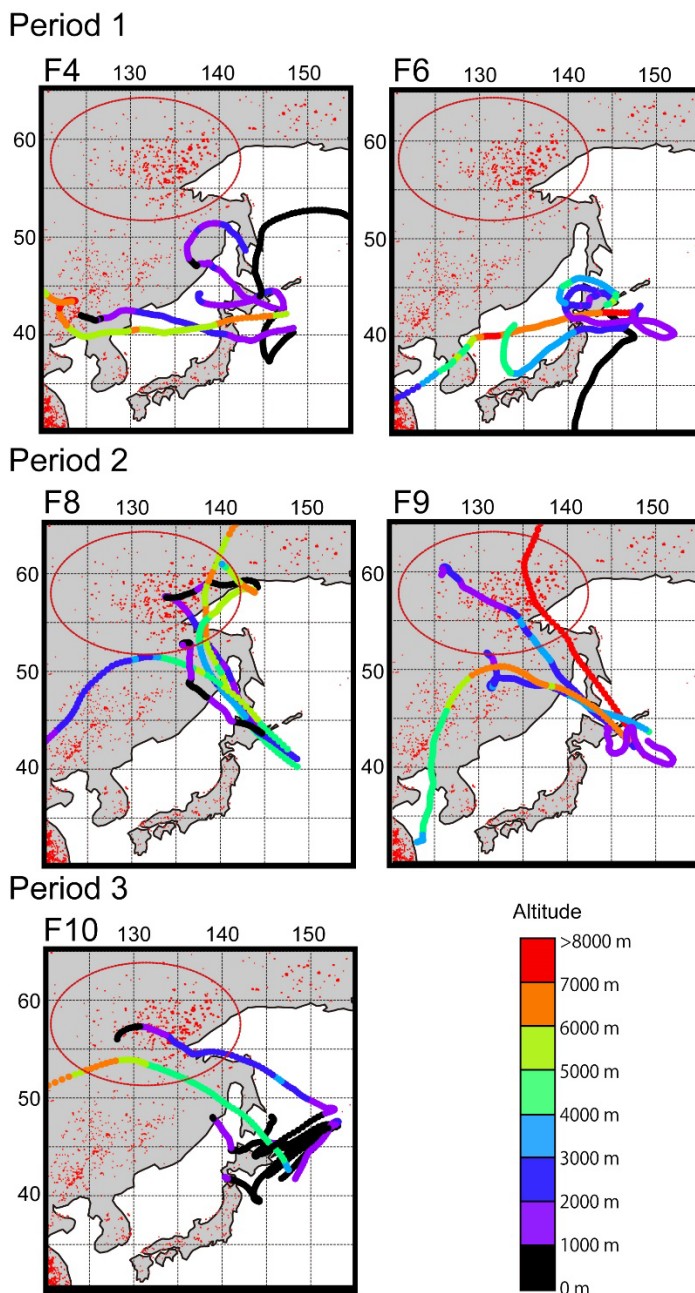

Figure 2. Back trajectories of sampled air parcels during each flight. Trajectories (120 h) started each hour along the flight passes. Colors along the trajectories indicate altitudes. Red dots on the maps indicate active fires during the sampling period (from July 22 to August 1, 2022) from NASA's Fire Information for Resource Management System (NASA FIRMS, 2025). The area of Siberian Forest biomass burning is marked with a red circle. The elevation plots of each trajectory and the plots

outside the selected area are shown in Figures S4 and S5, respectively.



## 3.2 Composition and mixing state of aerosol particles

In Figures 3 and 4, TEM images and elemental mappings show the mixing states of aerosol particles from simultaneously collected shipboard and airborne (< 1000 m) samples. The original TEM images indicate intact aerosol shapes collected on the substrate (left images in Figs. 3 and 4). The TEM images after STEM-EDS analysis reveal inclusions of, for example, soot and mineral dust within sulfate (middle images in Figs. 3 and 4). In these images, the electron beam removed beam-sensitive materials such as sulfate by exposing nearly one hour or ~ 100 repeated scans (Figs. 3 and 4) (Adachi et al., 2023; Egerton et al., 2004).

Organic materials, soot, and sea salts showed resistivity with an electron beam. In the elemental mapping images (right images in Figs. 3 and 4), the C distributions primarily represent organic materials and soot particles. Organic materials (secondary organic aerosols) typically coat sulfate and appear in a ring shape, as seen in samples from periods 2 and 3 (Fig. 3c-e). Soot particles have aggregate shapes and are coated or embedded in sulfate or organic matter (e.g., Fig. 4b). They appear as relatively bright spots in the C-mapping images. Na distributions mainly indicate the presence of sea salt and, to a lesser extent, mineral dust (e.g., Fig. 3b). In sea salt particles, Na coexists with S as sodium sulfate or Cl as sodium chloride. Sea salt particles are evident in samples from periods 1 and 3 and are almost negligible in samples from period 2. S distributions indicate the presence of sulfate with counter ions such as sodium, potassium, ammonium, or others. Since these sulfate-salt particles have crystal structure in the original TEM images, they looked dark and were decomposed after ~10 seconds of electron beam exposure (e.g., Figs. S1 and S2). K distributions mainly represent potassium sulfate, which is consist of S and K. K can also originate from mineral dust particles (Zhang et al., 2003). In the case of samples heavily affected by biomass burning (period 2), K and S also coexist with C, indicating that the potassium sulfate occurred with organic matter.

The dominant compositions and mixing states varied depending on the sampling periods. On the other hand, the airborne (< 1000 m) and shipboard samples from the same sampling periods showed similar compositions (Figs. 3 and 4), i.e., both airborne and shipboard samples showed dominant compositions in sea salt (sodium sulfate) (period 1), potassium sulfate with organics (period 2), and sea salt and potassium salt with organics (period 3). One difference, however, is that in Period 2, organics coated potassium sulfate in the airborne samples (Fig. 3c-d), while organics were homogeneously mixed with K and S in the shipboard samples (Fig. 4c-d).



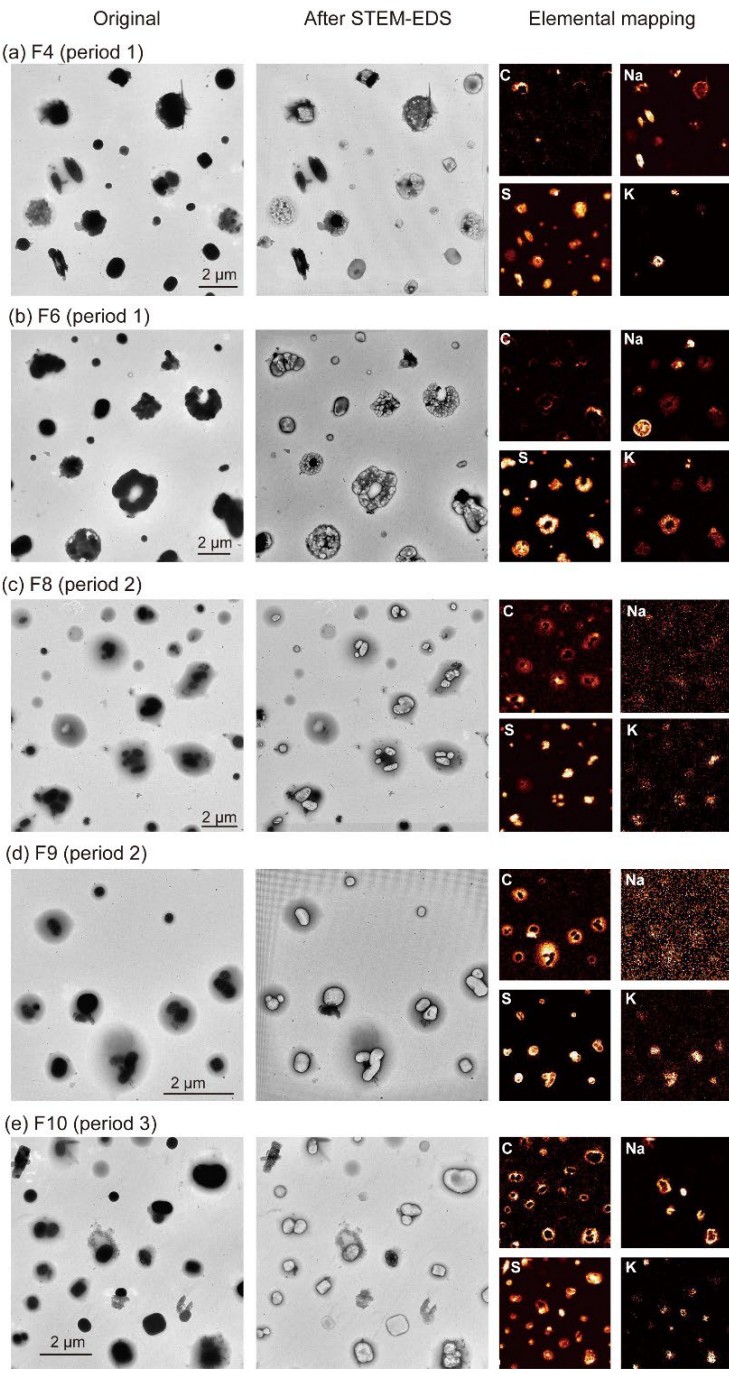

Figure 3. Mixing states and compositions of representative samples from each flight. Left: Original TEM images. Middle: TEM images after elemental mapping analyses. Sulfate and other beam sensitive materials were removed by the electron beam. Right: Elemental mappings of C, Na, S, and K. All samples were collected below 1000 m altitude. Sampling times were (a) July 22, 3:36-3:47; (b) July 27, 4:12-4:23; (c) July 29, 4:12-4:23; (d) July 30, 1:36-1:47; and (e) August 1, 1:48-1:59 UTC.





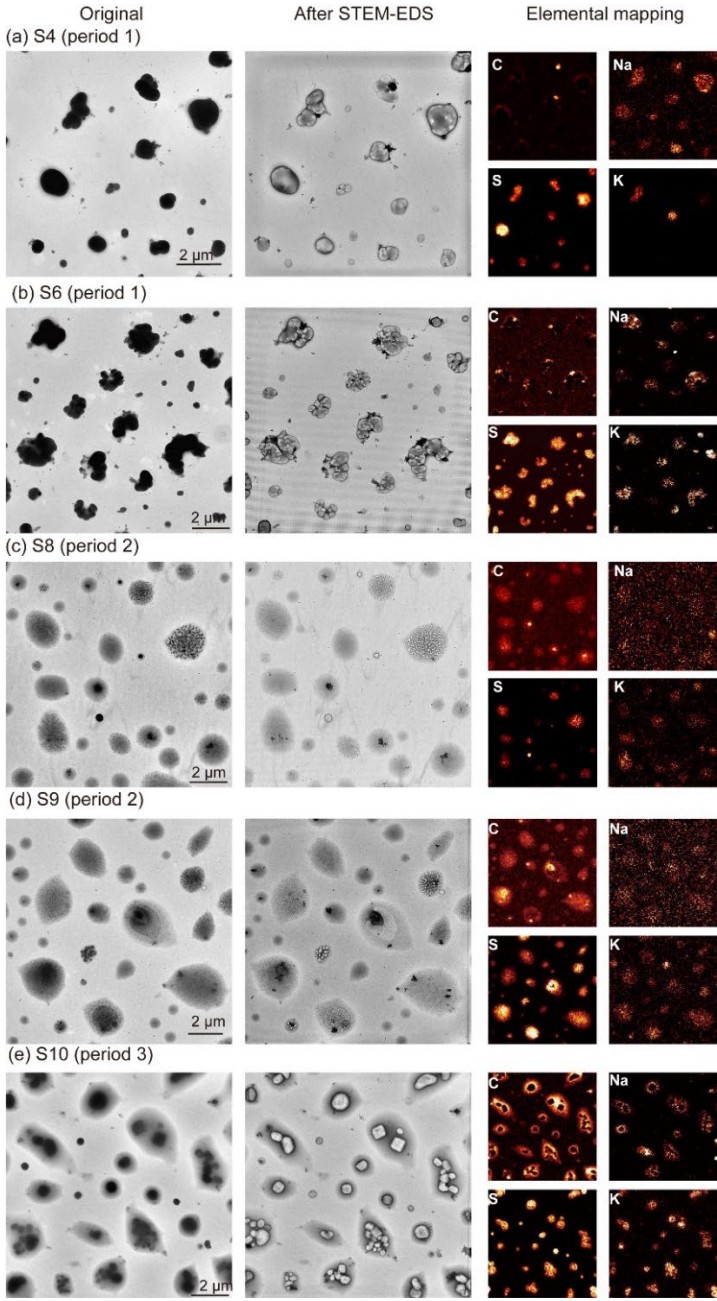

Figure 4. Mixing states and compositions of representative samples from shipboard samples (a)-(e): The numbers of shipboard samples (S) corresponding to those of airborne samples (F) with the same number. Left: Original TEM images. Middle: TEM images after elemental mapping analyses. Right: Elemental mappings of C, Na, S, and K. These samples were collected at 5:00 UTC (a-d) and 3:00 UTC (e) when the aircraft was over or near the ship.



### 3.3 Size-dependent particle compositions

We examined the size-dependent number fractions of seven aerosol types based on their compositions and area-equivalent diameters. Although some variations can be found in samples from different altitudes, the size-dependent number fractions are generally consistent within samples at < 3000 m from each period (Fig. S7). Here, we show the averaged size dependent number fractions of all samples from each flight to describe the particle abundance with size (Fig. 5).

The samples from period 1 (F4 and F6) showed that carbonaceous particle fractions are relatively higher in the smallest sizes (< 0.35 µm) (Fig. 5a-b). Sulfate fractions increased in the medium size range (0.35-1.0 µm). K-bearing particle fractions of F4 samples were not significantly different for all size ranges, whereas those of F6 slightly increased in larger sizes. Mineral dust particle number fractions increased in larger fractions. Sea salt fractions became higher in larger fractions and showed increases in the smallest fractions (<0.35 µm). In period 2, K-bearing fractions dominated in all sizes and increased in larger particle sizes (Fig. 5c-d). In contrast, sulfate and carbonaceous particle fractions increased in smaller fractions. There were few sea salt and mineral dust particles during the period 2. The fractions of period 3 were similar to those of period 1, but had larger fractions of sea salt and carbonaceous particles and fewer mineral dust particles than those of period 1 (Fig. 5e). The fractions from the shipboard measurements were mixtures of all sampling periods, i.e., K-bearing and sea salt fractions were higher in larger particles, except for the reduction of the sea salt fraction in the largest size bin (Fig. 5f), whereas the carbonaceous and sulfate fractions were higher in smaller particles.

In general, particle composition varies with size due to different formation processes and physical properties. For example, organic aerosols are rich in small particles (e.g., Jimenez et al., 2003; Moffet et al., 2008) when they grow through a new particle formation process. Sea salt particles, on the other hand, tend to be abundant in larger particle fractions due to their physical emission processes, e.g., breaking wave (Lewis and Schwartz 2004). K-bearing particles were commonly attached to other particles and have a relatively lower viscosity, which cause them to spread over the substrate (Adachi et al., 2025) and to be distributed in a wide size range and larger fractions.



Figure 5. Size-dependent number fractions of each flight and shipboard samples. The ranges of lognormal size bins are < 0.35,

0.35-0.50, 0.50-0.71, 0.71-1.0, 1.0-1.4, and >1.4 µm. (a)-(e) Airborne samples. (f) Average of all measured shipboard samples.





### 3.4 Altitude-dependent particle compositions

In addition to the sizes, the sampling altitude also influences the fractions of aerosol types (e.g., Liu et al., 2019). Here, we show the particle compositions as a function of sampling altitude (Fig. 6 and S8).

During periods 1 and 3, the sea salt fractions increased as the samples were collected at lower altitudes (Fig. 6 and S8). In addition, some increases in sea salt were observed in samples from > 7000 m, which could be due to long-range transport (e.g. the last sample in F10; Figs. S5 and S8). On the other hand, during period 2, the sea salt fractions did not largely differ depending on the altitude, but remained low. Sulfate and carbonaceous particle fractions generally increased at higher altitudes for all flights. The fractions of K-bearing particles decreased slightly at higher altitudes for samples from F4, F6, and F9. Small fractions of Ca-Mg-bearing particles were detected at high altitudes (~ >4000m) for all samples. The number fractions of mineral dust particles varied with the flights, increasing slightly at higher altitudes during F4 and F10. Overall, the particle fractions changed gradually with altitude, whereas particle mass concentrations varied widely with altitude (e.g., BC in Fig. S6).

In Fig. 7, we focus on aerosol compositions from sea spray and biomass burning sources within K-bearing particles. In figure 7a, we show the average values of the Na weight percent fractions over Na + K among K-bearing particles (Na / (Na + K)) along with their sampling altitudes. These ratios indicate how much sea salt components are mixed with biomass burning aerosols at the individual particle scale. In general, the ratios depend primarily on the sampling periods and secondarily on the altitudes. That is, during periods 1 (F4 and F6), Na had higher fractions than those of period 2 (F8 and F9). For period 3 (F10) samples, the fractions were high at lower altitudes but decreased with increasing altitude up to 2000 m. At altitudes > 5000 m, the Na fraction for periods 2 and 3 samples increased slightly (~0.2). The Na fractions from shipboard observations were similar to the airborne samples of periods 2 and 3 or were higher than those of period 1 at the lowest altitudes.

Biomass burning emits inorganic and organic matters, both of which mix within individual particles, i.e., organic matter condenses on or is mixed with sulfate (e.g., potassium sulfate), resulting in mixtures of sulfate and organic matter as shown in the TEM images (e.g., Fig. 3). The S weight percent fractions over S + C within the K-bearing particles (S / (S + C); S fraction) show the degree of the sulfate and organic mixing (Fig. 7b). Here, the S signals are from sulfate, and the C signals are from both substrate and organics. The C signals from the substrate do not vary much from particle to particle, while those from the organic matter vary depending on the degree of organic fraction, i.e. those with high organic fractions have lower S fraction values. Particles from periods 2 and 3 had lower S fraction values (higher C fractions) than those from period 1 in samples from lower altitudes (< ~4000 m). The results indicate that those from periods 2 and 3 had less sulfate and more organic matter from biomass burning, consistent with the mapping images.



Figure 6. Altitude-dependent number fractions of each flight. In (b) F6 and (d) F9, there are no TEM samples for 6000 m.



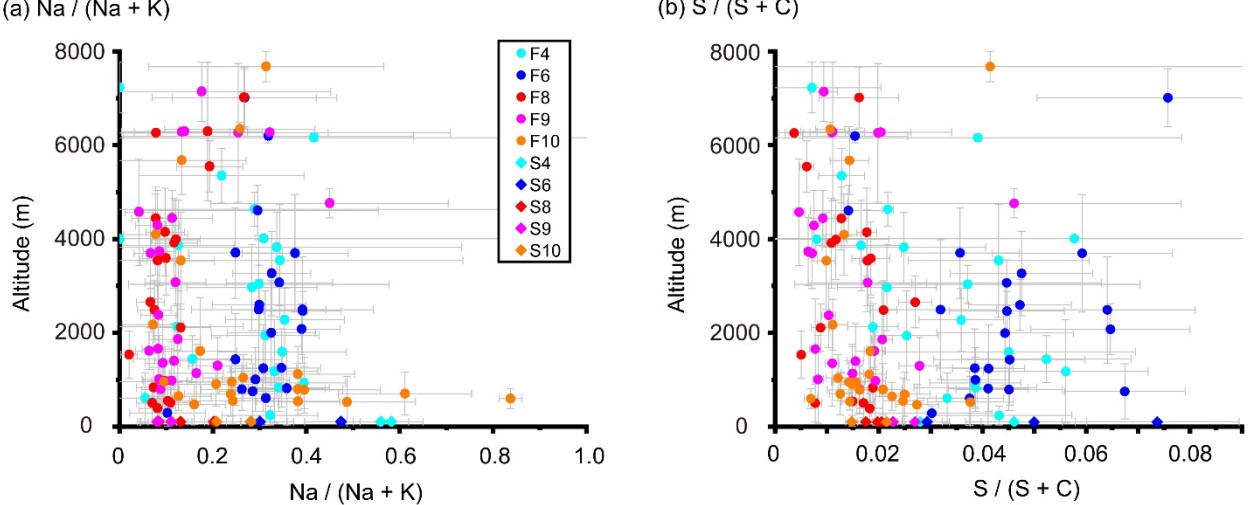

Figure 7. Ratios of tracer elements (Na-K and S-C) among K-bearing particles for all samples with different altitudes. (a) Na over Na + K (weight %) and (b) S over S + C (weight %). Circle and diamond symbols indicate airborne and shipboard samples, respectively. The ratios indicate sample averages. Error bars for the X-axis and Y-axis indicate 95% confidence intervals and ranges between the highest and lowest altitudes, respectively. F: Flight. S: Shipboard samples corresponding to airborne samples.

## 4. Discussions

### 4.1 Mineral dust and Ca-Mg-bearing particles

This study detected mineral dust particles defined by the presence of aluminosilicate (Al and Si). They were more abundant in larger particles (Fig. 5) and higher altitudes (Fig. 6) especially in the F4 samples. The F4 samples showed high fractions of mineral dust particles when the air parcel (4:00-5:00 UTC) originated from near the Gobi Desert (Fig. S5). Note that although the mineral dust number fractions were elevated in our measurements, a model study did not show a significant increase in mineral dust mass concentrations during F4 (Ohata et al., 2025). This discrepancy may be because the current study measured mineral dust number fractions in the TEM samples from the air parcel with low total aerosol mass concentrations, i.e., the atmospheric number concentration of the mineral dust particles can be lower than those in polluted periods such as period 2.

These mineral dust particles were found in aerosol samples without experiencing an apparent dust storm event (Ohata et al., 2025). The similar occurrences of mineral dust transport at 2-6 km over Japan in the absence of dust events have been reported by Matsuki et al. (2003). These results suggest that the mineral dust particles observed in the current study were occasionally transported at low concentrations regardless of apparent dust events. Mineral dust particles have potential climate impacts in several aspects. First, depending on temperature, mineral dust particles can act as ice nucleating particles and



influence ice cloud formation (Froyd et al., 2022; Kawai et al., 2021). Second, mineral dust can directly absorb and scatter light, affecting radiative forcing (Moteki et al., 2017; Satheesh and Krishnamoorthy, 2005). Lastly, when deposited in the

290 ocean, they become a nutrient for Fe, affecting the ocean carbon cycle (Adebiyi et al., 2023; Tagliabue et al., 2017).

In the current study, we specifically classified Ca- and Mg-bearing particles into an independent particle type (Ca-Mg-bearing particles) because of their unique occurrence, although their number fraction was small (0.2 % in total). They had the same Ca to Mg ratio (Fig. 8c), were rich in the finer fraction, and occurred at high altitudes (>4000 m). Their representative compositions suggest that they are a type of mineral dust, possibly dolomite ($CaMg(CO_3)_2$) mixed with other minerals (e.g.,

aluminosilicates). Song et al. (2005) found a relationship between Ca + Mg and $CO_3$ during airborne observations in an Asian outflow. Conny et al. (2019) detected dolomite particles in Hawaii from an Asian outflow sample. These studies suggest that these particles are common in Asian outflow aerosol samples. Furthermore, Li et al. (2007) have suggested that the source region of such dolomite in Asian dust is the northern edge of the Tibetan Plateau, including the Taklamakan Desert. Our findings are aligned with these previous results, i.e., they are abundant in high-altitude samples in the East Asia.

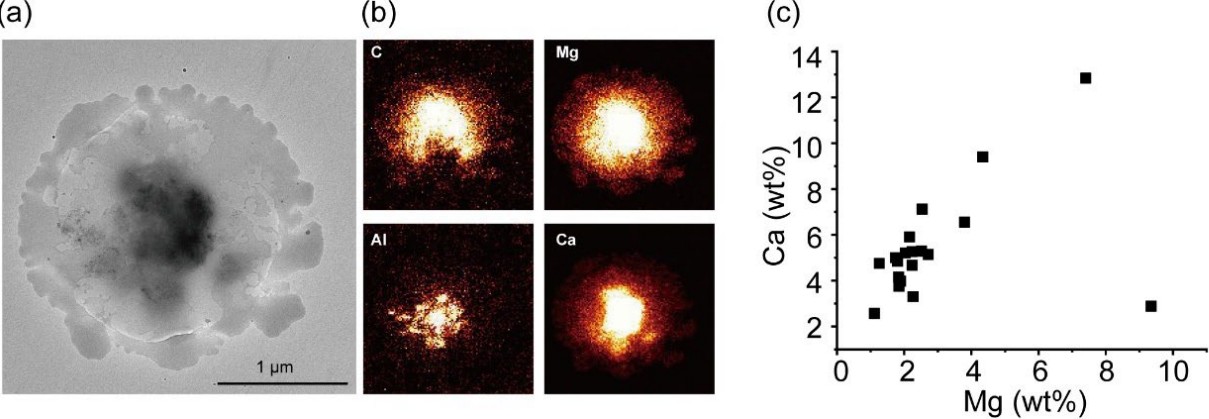

Figure 8. Shape and compositions of Ca-Mg-bearing particles. (a) TEM image. (b) Elemental mapping images for C, Mg, Al, and Ca. (c) Relation between Ca and Mg (weight %) for all Ca-Mg bearing particles (n=19).

## 4.2 Sources and compositions of aerosol particles during each period

Period 1 samples had relatively large sea spray influences due to their air parcel sources at low altitudes around the

305 ocean near Japan (Figs. 2 and S4). They contained high proportions of sea salt particles (Fig. 5) and of Na within K-bearing particles (Fig. 7a). The period 1 samples also contained certain amounts of K-bearing particles. Although they did not originate directly from the large Siberian biomass burning area, they could be from cropland biomass burning in northern China (Huang et al., 2024), regional haze from the large Siberian Forest biomass burning, or other sources emitting K-bearing particles (e.g., biofuel burning). Mineral dust particles from desert were also relatively abundant in this period as discussed in the previous

section. Substantial fractions of sulfate were observed during this period as well as other periods. Sulfate aerosols can originate from a variety of sources, including anthropogenic, marine, and volcanic, making them ubiquitous. We therefore interpret that





the number fractions of sulfate increases when other particles are scarce (e.g., high altitude samples). Overall, aerosol species from period 1 included sulfate, K-bearing, sea salt, and mineral dust. These aerosol species are generally consistent with those detected in the same area during other shipboard observation (Kawana et al., 2024), suggesting that the aerosols during period 1 are commonly observed in this region when there are no notable large pollution events.

Period 2 samples were influenced by the Siberian Forest biomass burning event. A pronounced increase in BC mass concentrations was observed by SP2 at altitudes between 1 and 3 km during this period (Fig. S6). On these occasions, the sampled air parcels were transported over the biomass burning area in the Siberian Forest (Fig. 2). The aerosol particle compositions were rich in organic matter (Fig. 3) and dominated by K-bearing fractions (Fig. 5), both of which are indicators of the influence of biomass burning (Andreae, 2019). High K-bearing particle fractions (0.4 - 0.8) were observed not only at altitudes of 1 - 3 km, where the BC enhancements were detected, but also at all observed altitudes, with the dominant fractions at < 6000 m in F8 or < 5000 m in F9 (Fig. 6). These altitudes included those with no clear increase in BC concentrations measured by SP2 (Fig. S6). These results suggest that, during period 2, the aerosol compositions over the observed area were affected by biomass burning at the individual particle scale, including those from air parcels with no apparent influences on BC mass concentrations. Siberian biomass burning plumes are known to travel long distances such as, to North America (Laing et al., 2016) and the Arctic (Warneke et al., 2009), indicating that they are a significant aerosol source globally. Sea salt particles, on the other hand, were rarely detected even near the sea surface during this period.

Period 3 samples showed a strong influence of sea spray as F10 mainly flew at low altitudes (~1000 m) (Fig. S8). The samples also showed organic coatings (Fig. 3e), possibly from biomass burning. The air parcels originated both from over the ocean (northwest of Japan) and from the biomass burning in Siberian Forest. Possible sources and compositions of the period 3 samples were mixtures of period 1 and 2.

**4.3 Comparison between airborne and shipboard observations**

The comparison of particle number fractions between airborne and shipboard observations in Figure 6 shows that the shipboard aerosol fractions are generally consistent with those of the low altitude (< 1000 m) airborne observations. Koike et al. (2025) showed that aerosol number concentrations from shipboard measurements were also consistent with those from airborne measurements. Our results show that they are generally comparable even for individual particle compositions. Differences, however, are that the shipboard measurements had slightly lower sea salt fractions than the corresponding airborne samples (< 1000 m) (Fig. 6) and that the sea salt fraction in the largest size bin (>1.4 μm) is smaller than that of the second largest bin (1.0-1.4 μm) in the shipboard samples (Fig. 5f), whereas all airborne samples have the largest sea salt fractions in the largest size bins (Figs. 5a-e). Although the exact reason for the discrepancy is unclear, the shipboard sampling was somewhat different from the airborne sampling in terms of sampling method, sampling time, and duration, and these differences may cause these discrepancies. Another difference is that the K-bearing particles in the airborne samples from period 2 had a core-shell structure (K-salt core with organic shell; Figs. 3c-d), whereas those in the shipboard samples had homogeneous mixtures of K and organic matter (Figs. 4c-d). One possible explanation is that the aerosol particles collected



from the shipboard observation were fully hydrated, as their homogeneous mixtures can form when both K salt and organic matter are in the liquid phase, although the exact process behind the difference remains to be determined.

## 4.4 Mixing states within individual particles

The influence of biomass burning has been detected in the western North Pacific during several shipboard observations (Taketani et al., 2025; Yoshizue et al., 2020). The current study further showed their vertical distributions at the
individual particle scale. Although we classified the particles separately as sea salt and K-bearing based on the presence of their tracer elements, they consisted of mixed compositions, especially for K-bearing particles (Fig. 7). The mixtures of sea spray and biomass burning aerosols have been observed near the coastal area in Thailand (Adachi et al., 2025) and over the Atlantic Ocean (Dang et al., 2022), where biomass burning and sea spray emissions are abundant. In the current study, we showed their mixtures from the sea surface to ~5000 m (Fig. 7). These results suggest that, although sea spray has a greater
influence at lower altitudes, its mixing proceeds vertically at the individual scale.

Riemer et al. (2019) showed two definitions related to particle mixing states: one is particle populations, which is the distribution of properties across the particles in the population, and the other is the single particles. In this study, the results of particle mixing states at the population scale are shown in Figure 6, and the results at the single particle scale are shown in Figure 7. This study demonstrated that the vertical aerosol variability occurs at both the particle population scale and the single
particle scale and showed their relationship, i.e., more sea salt or biomass burning particles in the particle population, more sea salt or biomass burning influence in the single particles.

## 5.  Summary and conclusion

This study analyzed individual aerosol particles collected from just above the sea surface to the atmosphere up to 8000 m by an aircraft and a research vessel over the western North Pacific in East Asia. The contributions from different
sources depended on the origin of the air parcel, such as ocean, desert, and biomass burning. Our results also revealed that particle compositions and mixing states varied with aerosol sizes and sampling altitudes. The current study showed important contributions from natural aerosols such as sea salt, mineral dust, and biomass burning particles, all of which were mixed within individual particles, and the extent of mixing varied with sampling periods and altitudes. The results of low-altitude airborne and shipboard samples were in reasonable agreement. This study highlighted a wide range of individual particle
compositions and evaluated how they were composed. Such knowledge of individual particle compositions and aerosol mixing states will contribute to a better assessment of aerosol contributions to climate.



**Data availability**

The shipboard measurement data (*R/V Shinsei-maru* KS-22-10 cruise) are available at https://www.jamstec.go.jp/datadoi/doi/10.17596/0003381 (JURCAOS and JAMSTEC, 2022). STEM-EDS data for all 375 individual particles and for the TEM sample average used in this study are available at https://doi.org/10.5281/zenodo.15178934 (Adachi, 2025).

**Author contributions**

KA conducted the TEM analysis and data processing. KA, TM, SO, and AY set up and executed the TEM sampling. KK conducted the back trajectory analysis. TM and NM measurements BC concentrations. MK and YK supervised the aircraft 380 and the research vessel observations, respectively. KA prepared the manuscript with contributions from all coauthors.

**Competing interests**

The authors declare that they have no conflict of interest.

**Acknowledgement**

We are indebted to all participants of A-Force 2022 for their cooperation and support, especially to Mr. Ohmi and 385 Mr. Saito (U. Tokyo) for collecting TEM samples on the aircraft. The authors also thank the pilots and flight staff of Diamond Air Service Inc. and the crew of *R/V Shinsei-maru* for their support. We acknowledge the use of imagery from NASA's Fire Information for Resource Management System (FIRMS) (https://earthdata.nasa.gov/firms), part of NASA's Earth Observing System Data and Information System (EOSDIS) and NIES for the back trajectory model (https://db.cger.nies.go.jp/ged/metex/ja/index.html).

**Financial support.**

This research has been supported by the Environmental Research and Technology Development Fund (JPMEERF20232001) of the Environmental Restoration and Conservation Agency of Japan, the Global Environmental Research Coordination System from the Ministry of the Environment of Japan (MLIT1753 and MLIT2253), the Arctic Challenge for Sustainability II (ArCS II) (JPMXD1420318865), and the Japan Society for the Promotion of Science (JSPS) 395 KAKENHI program (grant numbers JP19K21905, JP19H04259, JP19H05699, JP19H05700, JP23H03531, JP23K28210, JP24H00761, JP23KJ2144, and JP23K28221).



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
