# Peer review of "Individual particle compositions and aerosol mixing states at different altitudes over the ocean in East Asia"

_EGUsphere, 2025_

## Author Comment (AC1)

**Reviewer comments are shown in Bold.**

Author replies are shown in normal font.

*Revised texts are shown in Italic.*

**Reviewer 1**

**This manuscript presents a comprehensive analysis of individual aerosol particle**

**compositions and mixing states collected over the western North Pacific during**

**summer 2022, utilizing coordinated aircraft and research vessel measurements. The**

**authors employed transmission electron microscopy with energy dispersive X-ray**

**spectrometry (TEM-EDS) to characterize aerosol particles from sea surface to**

**approximately 8000 m altitude. The study identifies three distinct periods based on air**

**mass origins and demonstrates how particle compositions vary with size, altitude, and**

**source regions, providing valuable insights into aerosol mixing states at the individual**

**particle level. However, several aspects of the manuscript require improvement, as**

**detailed below.**

Authors' comments: We appreciate the reviewer reading our work and providing constructive comments.

**R1-1: The Abstract should more precisely articulate the major findings and conclusions**

**of the study.**

A1-1: The abstract has been revised. It now describes the precise findings of this study.

**R1-2: The introduction should better emphasize the significance of aerosol particle**

**composition, size distribution, and mixing state in climate impacts.**

A1-2: We added sentences to emphasize the suggested points.

*1. Introduction: Such changes in the physical, chemical, and optical properties of particles*

*due to their mixing states affect the radiative forcing of aerosols, thereby altering their*

*response to climate (Adachi et al., 2010; Cappa et al., 2012; Chung and Seinfeld, 2002;*

*Jacobson, 2001). For example, light-absorbing particles (e.g., black carbon) coated with*

*light-scattering materials (e.g., sulfate and organic matter) enhance their light absorption by*

*focusing incident light on the light-absorbing materials (Bond et al., 2013; Wang et al., 2025).*

*Mixing of hygroscopic materials with hydrophobic particles can alter their hygroscopicity,*

*influencing their CCN and INP activities (Riemer et al., 2019; Lohmann et al., 2020). Mixing*

*aerosols with other materials can alter their size distribution, which influences their optical*

*properties (Moteki et al., 2007).*

**R1-3: The introduction and discussion sections would benefit from more detailed**

**information and comparative discussion of aerosols at different altitudes, including**

**specific quantitative values of their influence, which would enhance the paper's impact.**

A1-3: We have added a sentence about altitude-dependent aerosol compositions to the

Introduction. We have also included a discussion of sea salt fractions at high altitudes.

*1. Introduction: Information about altitude-dependent aerosol compositions is especially*

*important for evaluating, for example, their influence on health at ground level and on CCN*

*activity at cloud height.*

*Section 3.4: As the samples were collected at lower altitudes during periods 1 and 3, the sea*

*salt fractions increased (Fig. 6 and S9), which is consistent with the sea salt observations*

*over the global troposphere (Murphy et al., 2019). In addition, some increases in sea salt*

*were observed in samples from > 7000 m (e.g. the last sample in F10; Figs. S5 and S9). The*

*sea salt particles at high altitudes possibly originated from different regions than those at*

*low altitudes because they had different air parcel history (Figs. 2 and S4) and size*

*distribution (Fig. S8), although the specific origin and transport path of the sea salt particles*

*at high altitudes remain unclear. On the other hand, during period 2, the sea salt fractions*

*did not largely differ depending on the altitude, but remained low.*

**R1-4: Regarding Figure 3d and 4c-d: The elemental mapping images show ubiquitous**

**weak Na signals covering almost the entire mapping area. Is this Na derived from**

**combustion processes?**

A1-4: We think that the Na originated from both sea spray and combustion processes.

Although these samples were dominated by biomass burning aerosols (K-bearing particles), they still included sea salt particles (Fig. 6). Thus, although the Na contributions from sea spray was weaker than in other periods, the samples could still contain Na from sea spray. Na contributions from biomass burning (i.e., combustion process) is discussed in A2-4 and supplementary figure 10. We added some explanations to the revision.

*Section 3.2: Sea salt particles contain Na as sodium sulfate or possibly sodium chloride, as*

*shown in samples from periods 1 and 3 (Figs. 3 and 4). Samples from period 2 contain much*

*smaller amounts of Na than samples from periods 1 and 3, though trace amounts of Na may*

*still be present, possibly originating from sea spray and biomass burning (Adachi et al., 2025).*

**R1-5: In the supplementary materials, particles are classified as K-bearing when K >**

**0.01 wt%. Did you perform EDS analysis on every individual particle in the samples?**

A1-5: Using STEM-EDS, we measured all particles within the selected area of 6,000×

(10,708 particles from 122 TEM grids and 89 particles on average) (Section 2.3). In other words, we performed EDS analysis on every particle we measured, i.e., yes, for the question.

The threshold of 0.01 wt% indicates that all particles containing K (excluding mineral dust and sea salt) were identified as K-bearing. Please note that, while reviewing the data for this revision, we recognized that the K-bearing particle with the lowest wt% in this category is

0.028 wt%. Thus, we revised the threshold to 0.02 wt%.

[Figure]

**R1-6: Lines 316-325: When discussing BC between 1 and 3 km, it is essential to elaborate**

**on the climate implications, particularly the effects at higher altitudes (Lohmann et al.,**

**2020; Wang et al., 2025).**

A1-6: We revised the relevant sentences and added these references.

*1. Introduction: For example, light-absorbing particles (e.g., black carbon) coated with light-*
*scattering materials (e.g., sulfate and organic matter) enhance their light absorption by*
*focusing incident light on the light-absorbing materials (Bond et al., 2013; Wang et al., 2025).*
*Mixing of hygroscopic materials with hydrophobic particles can alter their hygroscopicity,*
*influencing their CCN and INP activities (Riemer et al., 2019; Lohmann et al., 2020).*

**R1-7: Line 240 states: "In addition, some increases in sea salt were observed in samples**
**from > 7000 m, which could be due to long-range transport." This explanation lacks**
**clarity. What specific mechanisms of long-range transport would increase sea salt**
**concentrations above 7000 m? Are sea salt particles more susceptible to high-altitude**
**transport compared to other aerosol types?**

A1-7: The sea salt particles at high altitudes possibly originated from different regions than
those at low altitudes because they had different air parcel history (Figs. 2 and S4) and size
distribution (Fig. S8). Generally, a low-pressure system over the ocean may have uplifted the
sea salt particles, but their specific sources, mechanisms, and transport path remained unclear.
We explained these aspects in the revised text.

*Section 3.4: As the samples were collected at lower altitudes during periods 1 and 3, the sea*
*salt fractions increased (Fig. 6 and S9), which is consistent with the sea salt observations*
*over the global troposphere (Murphy et al., 2019). In addition, some increases in sea salt*
*were observed in samples from > 7000 m (e.g. the last sample in F10; Figs. S5 and S9). The*
*sea salt particles at high altitudes possibly originated from different regions than those at*
*low altitudes because they had different air parcel history (Figs. 2 and S4) and size*
*distribution (Fig. S8), although the specific origin and transport path of the sea salt particles*
*at high altitudes remain unclear. On the other hand, during period 2, the sea salt fractions*
*did not largely differ depending on the altitude, but remained low.*

**R1-8: The manuscript would benefit from more comprehensive comparison with**
**previous single-particle studies in the region. How do the observed mixing states**
**compare with other studies over the North Pacific? Are the biomass burning signatures**
**consistent with other Siberian fire events? How do the mineral dust characteristics**
**compare with known Asian dust compositions?**

A1-8: We added the following discussions to address the reviewer comments.

*Section 4.1: This configuration of an insoluble aluminosilicate core with a Ca-Mg-rich coating is similar to that described by Tobo et al. (2010). They proposed that Ca-rich particles transformed into aqueous droplets. Our particles also appear to be in the aqueous phase when collected because they spread over the substrate.*

*Section 4.2: Previous shipboard measurements in this region have also observed the influence of Siberian biomass burning but with more organic aerosol particles including tarballs (Yoshizue et al., 2020), whereas the current study did not detect tarballs. A possible explanation for this discrepancy is different fire conditions, i.e., biomass burning with flaming phases emit more K and BC, while biomass burning with smoldering phases emits more organic matter and tarballs (Adachi et al., 2024).*

**R1-9: As mentioned earlier, while addressed in the conclusions, the climate implications could be expanded to include: specific discussion of how the observed mixing states affect optical properties; implications for ice nucleation and cloud condensation nuclei activity; and relevance for model parameterizations.**

A1-9: As shown in A1-2, we added climatological implications. Additionally, we enhanced the relevant sentences in the revised text.

*1. Introduction: For example, light-absorbing particles (e.g., black carbon) coated with light-scattering materials (e.g., sulfate and organic matter) enhance their light absorption by focusing incident light on the light-absorbing materials (Bond et al., 2013; Wang et al., 2025). Mixing of hygroscopic materials with hydrophobic particles can alter their hygroscopicity, influencing their CCN and INP activities (Riemer et al., 2019; Lohmann et al., 2020). Mixing aerosols with other materials can alter their size distribution, which influences their optical properties (Moteki et al., 2007).*

*5. Summary and conclusion: The compositions, mixing states, and sizes of aerosols directly influence their optical properties as well as their CCN and INP activities, depending on the sampling altitude.*

*5. Summary and conclusion: Such knowledge of individual particle compositions and mixing states will improve model parameterizations, resulting in a more accurate assessment of the aerosol contributions to the climate.*

**Reviewer 2**

**Adachi et al. presented a comprehensive single particle analysis of aerosols collected at different altitudes over the western North Pacific Ocean. This work provides valuable insights into the source and composition of aerosols in East Asia, which are not yet fully understood. Overall, this study is well-designed, and the paper is well-written. There are some places where I have some comments. Please see my comments below.**

Authors' comments: We appreciate the reviewer reading our work and providing constructive and supportive comments.

**General comments:**

**R2-1: I have some questions about the particle classification matrix. First, how did you develop the matrix? Based on the literature or K-means cluster? Secondly, it seems like the classification can lead to conditions where particles can be classified into multiple classes (e.g., a particle with Al > 0.5 wt%, Si > 2 wt%, Na >1 wt%, and Mg > 0.01 wt% can be classified into both mineral dust and sea salt classes). Could you double-check your flow chart? Is that because you did not list the criteria to exclude particles from other classes (e.g., mineral dust should be Al > 0.5 wt% and Si > 2 wt% and Na <1 wt%, Mg < 0.01 wt%, K < 0.01 wt%, Ca < 0.5 wt%, S < 1 wt%, C+O < 90 wt%)? If yes, please add a note to make this clear to audiences.**

A2-1: Similar flowcharts have been used in our previous papers (e.g., Adachi et al., 2021, 2022, 2023, 2025), but the tracer elements and threshold values were modified based on the dominant aerosol sources. For example, the current study included Ca-Mg-bearing particles because they were uniquely found in the current samples. We revised the classification method in Section 2.4.

*Section 2.4: Similar flowcharts have been used in our previous papers (e.g., Adachi et al., 2021, 2022, 2023, 2025), but the tracer elements and threshold values were modified based on the dominant aerosol sources. For example, the current study included Ca-Mg-bearing particles because they were uniquely found in the current samples.*

Particles are classified into one category followed by the flow in Fig S3. For example, the particle mentioned by the reviewer 2 (**a particle with Al > 0.5 wt%, Si > 2 wt%, Na >1 wt%, and Mg > 0.01 wt%**) is classified as a mineral dust particle; i.e., the flow chart goes to the next step only if the above criteria is NO. Nearly all ambient aerosol particles are mixtures of several components, especially secondary materials, as shown in Figs. 3 and 4.

Thus, we classified the particles focusing mainly on primary aerosol particles. We explained this as follows:

*Section 2.4: Although individual particles are mixtures of several components (e.g., sea salt*

*+ sulfate particles), we have simply classified them into these single categories, focusing*

*mainly on primary aerosol particles and prioritizing the smaller particle type numbers (1 to*

*7) shown in the above paragraph. Such prioritized aerosol species include mineral dust and*

*sea salt particles, which are important INP and CCN contributors to the activation of ice*

*crystals and water droplets, respectively.*

[Figure]

**R2-2: Section 2.6 needs to provide more information to help people reproduce the**

**results.**

A2-2: We listed the starting points for each trajectory (Table S1) and additional information in the text, caption, and Data availability section.

*Section 2.6: The starting points of these back-trajectories were based on the altitude and*
*position of the aircraft every hour from 2:00 to 6:00 UTC (Table S1).*

*Caption of Fig. 2: Figure 2. Back trajectories of sampled air parcels during each flight.*
*Trajectories (120 h) started each hour along the flight passes. Colors along the trajectories*
*indicate altitudes. Red dots on the maps indicate active fires during the sampling period (from*
*July 22 to August 1, 2022) from NASA's Fire Information for Resource Management System*
*(NASA FIRMS, 2025). The area of Siberian Forest biomass burning is marked with a red*
*circle. The elevation plots of each trajectory and the plots outside the selected area are shown*
*in Figures S4 and S5, respectively.*

*Data availability section: The METEX back-trajectory model provided by NIES is available*
*at https://db.cger.nies.go.jp/ged/metex/en/index.html.*

*Table S1. The positions of the aircraft at each hour.*

| | UTC | Long (deg) | Lat (deg) | Height (m agl) |
|---|---|---|---|---|
| F4 | 0200 | 146.5589 | 42.5128 | 1732.6 |
| | 0300 | 148.0219 | 40.1481 | 134.9 |
| | 0400 | 148.0071 | 40.6131 | 1626.8 |
| | 0500 | 147.1833 | 42.0905 | 5864.5 |
| | 0600 | 144.16 | 43.8796 | 66.7 |
| F6 | 0200 | 144.5813 | 42.0304 | 177.3 |
| | 0300 | 146.6322 | 42.2479 | 4922.2 |
| | 0400 | 147.9427 | 42.954 | 1143 |
| | 0500 | 147.1251 | 42.3443 | 209.6 |
| | 0600 | 144.4347 | 43.7994 | 968.9 |
| F8 | 0200 | 147.0213 | 42.2252 | 1159.4 |
| | 0300 | 148.482 | 40.4313 | 4268.9 |
| | 0400 | 148.4498 | 41.2466 | 2586.1 |
| | 0500 | 147.3599 | 42.2196 | 4924.7 |
| | 0600 | 144.455 | 43.8511 | 809.2 |
| F9 | 0200 | 147.2814 | 41.9291 | 2685.3 |
| | 0300 | 148.5012 | 41.0827 | 4982.4 |
| | 0400 | 148.9798 | 43.5075 | 2184.6 |
| | 0500 | 146.6079 | 42.6057 | 6280.7 |
| | 0600 | 144.16 | 43.8797 | 66.5 |
| F10 | 0200 | 145.0106 | 41.6363 | 531 |
| | 0300 | 146.5894 | 41.3135 | 634.3 |
| | 0400 | 148.1675 | 41.2888 | 1194.9 |
| | 0500 | 147.3906 | 42.2311 | 3969.3 |
| | 0600 | 144.166 | 43.8654 | 117.4 |

**R2-3: I suggest adding some labels in the TEM images to indicate different types of particles.**

A2-3: A figure showing the aerosol species in Fig. 3 was added in Supplementary information (Fig. S7).

*Section 3.2: The TEM images after STEM-EDS analysis reveal inclusions of, for example, soot and mineral dust within sulfate (middle images in Figs. 3 and 4), which are also indicated in Fig. S7.*

[Figure]

Figure S7. Possible aerosol species in Fig. 3 are indicated by arrows.

**R2-4: L 249-250, "In figure 7a …. Along with their sampling altitudes." I am not sure Na/(Na+K) is a good proxy for sea salt mixed with biomass burning aerosol, since biomass burning also emits trace amounts of Na, and sea salt also contains some K. The K salt typically has lower solubility than Na salt, so they will crystallize first before Na salt and from individual K salt particles. Could you please add some discussion about this?**

A2-4: The possible contribution of Na from biomass burning was discussed in our previous paper (Adachi et al., 2025). In that paper, non biomass burning Na was estimated as [non-biomass burning Na] = [Na] – [K] × 0.06. In the plots, values of [non-biomass burning Na] are slightly smaller than those of [Na] without adjustment, but both figures are essentially the same. Since the estimate is based on an assumption and the both figures do not change the discussion, we would keep the original figure and show the [non-biomass burning] plots as a supplementary figure S10 for reference.

The crystallization of K and Na salts is interesting points. In fig. 4e, some particles have K-salt core with Na-salt surroundings. We added a discussion about it.

*Section 3.2: Potassium salts (e.g., potassium sulfate) typically have lower solubility than*
*sodium salts (e.g., sodium sulfate) and may crystallize at the ambient temperatures (~25℃)*
*before sodium salts do, resulting in the formation of a potassium salt core surrounded by*
*sodium salt (Period 3 in Figs. 3e and 4e).*

[Figure]

*Figure S10. Comparison of the sodium fraction with consideration of the sodium*
*contributions from biomass burning. This figure is the same as Fig. 7a but with modified*
*sodium contributions from biomass burning based on an assumption by Adachi et al. (2025):*
*[$Na_{non-biomass\ burning}$] = [Na] – [K]×0.06.*

**R2-5: Could you add Si maps to help identify the mineral dusts? It will be useful to**
**support Figure 8 particle is a dust since Ca, Mg, and CO3 can exist in marine**

A2-5: Si map was added. We also extended our discussion in the relevant paragraph.

[Figure]

Figure 8. Shape and compositions of Ca-Mg-bearing particles. (a) TEM image. (b) Elemental
mapping images for C, Mg, Al, Si, and Ca. (c) Relation between Ca and Mg (weight %) for
all Ca-Mg bearing particles (n=19).

Section 4.1: This configuration of an insoluble aluminosilicate core with a Ca-Mg-rich
coating is similar to that described by Tobo et al. (2010). They proposed that Ca-rich
particles transformed into aqueous droplets. Our particles also appear to be in the aqueous
phase when collected because they spread over the substrate.

**Specific comments:**

**R2-6: L100-103, "Second, we selected … (STEM-EDS)." Could you explain a little bit
more about the criteria you used to select the representative area? Moreover, does
"6,000 X" mean magnification of 6,000?**

A2-6: Yes, for the last question. We revised the relevant sentences.

Section 2.3: Second, based on analyses of the TEM images (Figs. S1 and S2), we selected
one representative area per TEM sample with an adequate number of particles (i.e., not
overloaded). We measured TEM samples for the five flights (109 TEM samples) and 13
shipboard TEM samples that covered the synchronized measurements with the aircraft (two
or three shipboard samples during each flight) (Table 1). Third, we measured all particles
(89 particles on average) within the selected area with a magnification of 6,000 (~220 $\mu m^2$)
using STEM with EDS measurements (STEM-EDS).

**R2-7: L111-112, "Elemental compositions … aquation times." It seems that your**
**particle classification did not use all elements (e.g., Ti to Pb were not used). Should you**
**remove them and renormalize the wt%?**

A2-7: We selected these elements (e.g., Ti and Pb) in case if there were strong Asian dust or
anthropogenic pollution events. After careful analysis, we found that these elements were
rarely detected and were not used further analyses. However, we do not want to exclude them
from the measurements because they still provide information of the particle compositions.
Please note that excluding these elements from the STEM-EDS measurements would require
recalculating all particles because all selected elements were used to estimate the self-
absorption of X-rays.

**R2-8: L260-262, "The C signals … S fraction values." I understand that TEM uses a**
**very high accelerating voltage, which enables electrons to easily penetrate the entire**
**particles. However, it might still be beneficial to show the calculation of the**
**penetration depth for pure metal or add a citation to make readers understand this**
**concept.**

A2-8: We revised the sentence and added a reference.

*Section 3.2: The C signals from the substrate do not vary much from particle to particle*
*because the electron beam easily penetrates the entire particle and substrate (Geng et al.,*
*2010), while those from the organic matter vary depending on the degree of organic*
*fraction, i.e. those with high organic fractions have lower S fraction values.*

**Reviewer 3**

**This manuscript presented the elemental composition and mixing state of aerosol**
**particles collected in the North Pacific at different altitudes. It is important to gain a**
**better understanding of particle mixing state at the single-particle level, as this will**
**affect radiative forcing and cloud formation. The manuscript found different impacts**
**from different sources, including biomass burning, marine emission, and deserts**
**during the field campaign. This study provides a valuable data set on the vertical**
**profile of particle composition and mixing state. However, there are several issues that**
**need to be addressed before it can be considered for publication.**

Authors' comments: We appreciate the reviewer reading our work and providing
supportive and helpful comments.

**General comments:**

**R3-1: Please revise the abstract to focus more on the main results of this investigation.**
**The definition of different sampling periods in the abstract is unclear. The conclusions**
**in Line 21-25 are too general and could apply to many similar studies.**

A3-1: The abstract has been revised. It now describes the definition of these sampling periods.
The conclusion now focuses on the precise findings of the study.

*Abstract: The samples were classified into three periods based on the sampled air parcel*
*sources: ocean and desert (period 1), Siberian Forest biomass burning event (period 2), and*
*their mixtures (period 3).*

*Abstract: During periods 1 and 3, the sea salt fractions increased as the samples were*
*collected at lower altitudes. The compositions of biomass burning and sea spray were mixed*
*at individual particles, with higher fractions of Na and K during period 1 and period 2,*
*respectively, than in other periods. Our analysis of individual particle analysis revealed a*
*wide range of compositions and mixing states of particles, which depend on the aerosol*
*source, size, and altitude.*

**R3-2: As the authors also mentioned, the method for particle classification is**
**simplified (Line 120-132). Since you have the relative elemental composition of**
**individual particles, is there a better classification scheme or clustering method that**
**could distinguish particles with mixed components, which may help to provide a better**
**description on the mixing state of particles.**

A3-2: We appreciate the reviewer's suggestion, and we aim to demonstrate their complex
mixing states directly using TEM images and various elemental mapping images (e.g.,
Figures 3, 4, and 8 and supplementary figure 7). In figures 5 and 6, we attempt to categorize
complexly mixed particles as simply as possible so that readers can easily identify the
contributions of each major aerosol type at different sizes and altitudes. Additionally, we
provide individual particle composition data in the data archive, which allows interested
individuals to access and analyze the raw particle composition data. While there are many
ways to show the mixing states of particles as the reviewer suggests, we hope this strategy is
helpful to readers to understand the mixing states of our samples.

**R3-3: The discussion on the mixing state of the particle population and individual**
**particles is not thorough, especially at the individual particle level. In both cases, the**

**mixing state matrix mentioned by Riemer et al (2019) would provide more**
**quantitative results to describe the mixing state.**

A3-3: We agree that the concept of particle mixing states, i.e., particle populations and single
particles, as described by Riemer et al. (2019), is useful. Thus, we briefly discussed them in
Section 4.4. In this section, we demonstrate that our results exemplify such mixing states in
an ambient aerosol particle. We provided all particle information in the archived data, which
may be helpful for reproducing a particle-resolved mixing state model. While a further
analysis of the numerical analysis of mixing states based on the current dataset would be
interesting and worthwhile for a future study, it would expand the scope of this paper too
much. Therefore, we prefer to focus on the topics currently discussed in this paper.

R3-4: **Na-containing particles can also be emitted from biomass burning affect the ratio.**
**Carbon in the EDS is semiquantitative. Secondary organic aerosol or materials would**
**also contribute to the carbon in the particles. These should be considered.**

A3-4: Na from biomass burning was discussed in the revised text and in Figure S10.

*Section 3.4: In addition to sea spray, small fractions of Na may originate from biomass*
*burning. Based on the assumption used in our previous study (Adachi et al., 2025), we*
*evaluated the influence of Na from biomass burning emissions. The results showed that the*
*presence or absence of Na emissions from biomass burning had a negligible effect on the*
*current discussion of Na fractions (Fig. S10).*

Limitation of carbon measurements and the contribution of secondary organic aerosol are
discussed in the revised paragraph.

*Section 3.4: Biomass burning emits inorganic and organic matters, both of which mix within*
*individual particles, i.e., secondary organic matter condenses on or is mixed with sulfate (e.g.,*
*potassium sulfate), resulting in mixtures of sulfate and organic matter as shown in the TEM*
*images (e.g., Fig. 3). Although absolute quantification of C weight percent from organic*
*matter is difficult due to substrate interference and the varying thickness of organic matter,*
*S weight percent fractions over S + C within K-bearing particles (S/S+C; S fraction) can*
*indicate the degree of sulfate and organic matter mixing (Fig. 7b). Here, the S signals are*
*from sulfate, and the C signals are from both substrate and organics. The C signals from the*
*substrate do not vary much from particle to particle because the electron beam easily*
*penetrates the entire particle and substrate (Geng et al., 2010), while those from the organic*

*matter vary depending on the degree of organic fraction, i.e. those with high organic fractions*

*have lower S fraction values.*

**Specific comments:**

**R3-5: Line 19, particle composition can't be determined only by TEM but with EDS.**

A3-5: Revised

*Abstract: Measurements of particle composition using transmission electron microscopy with energy-dispersive X-ray spectrometry revealed that samples from period 1 had high sea salt and mineral dust fractions, while samples from period 2 had high fractions of potassium-bearing particles with organics and black carbon.*

**R3-6: Line 230-232, please elaborate on this statement.**

A3-6: We provided more explanation in the relevant sentences.

*Section 3.3: K-bearing particles commonly attach to other particles across a wide size range, making them ubiquitous within all size bins (Figs. 5a, 5b, and 5e). Additionally, K-bearing particles have relatively low viscosity, causing them to spread over the substrate (Adachi et al., 2025) and be distributed in larger size bins (Figs. 5c, 5d, and 5f).*

**R3-7: Line 329, organic coating may come from Secondary organic materials.**

A3-7: Yes, most of the organic coatings measured in the current samples were secondary organic materials. We revised the sentence.

*Section 4.2: The samples also showed organic coatings of secondary organic matter (Fig. 3e), possibly from biomass burning.*

**R3-8: Line 345, The dehydration RH for the $K_2SO_4$ is very high. Is there any RH data to support this?**

A3-8: Yes. The RH data on the *R/V Shinsei-maru* is available at the data archive site (JURCAOS and JAMSTEC, 2022) and is plotted below. We included the RH data range as a text in the revised paper.

*Section 4.3: One possible explanation is that the aerosol particles collected from the shipboard*
*observation were fully hydrated due to the high relative humidity (RH) ranging from 75% to 100%*
*throughout the entire cruise (JURCAOS and JAMSTEC, 2022). The RH values exceeded the*
*deliquescence and efflorescence RH values of potassium salts (e.g., 97% and 60% for K2SO4,*
*respectively; Freney et al., 2009), suggesting that they were hydrated when collected, i.e., these*
*homogeneous mixtures can form when both the K salt and the organic matter are in the liquid phase.*

[Figure]

*Figure. Plots of relative humidity on the R/V Shinsei-maru during the entire cruise (JURCAOS and*
*JAMSTEC, 2022).*